# Structure-based discovery of potent and selective melatonin receptor agonists

**Nilkanth Patel[1], Xi Ping Huang[2,3], Jessica M Grandner[1†], Linda C Johansson[1], Benjamin Stauch[1], John D McCorvy[2,3‡], Yongfeng Liu[2,3], Bryan Roth[2,3,4], Vsevolod Katritch[1]\***

[1]Department of Biological Sciences and Department of Chemistry, Bridge Institute, USC Michelson Center for Convergent Biosciences, University of Southern California, Los Angeles, United States; [2]Department of Pharmacology, University of North Carolina Chapel Hill Medical School, Chapel Hill, United States; [3]National Institute of Mental Health Psychoactive Drug Screening Program, Department of Pharmacology, University of North Carolina Chapel Hill Medical School, Chapel Hill, United States; [4]Division of Chemical Biology and Medicinal Chemistry, University of North Carolina Chapel Hill Medical School, Chapel Hill, United States

**Abstract** Melatonin receptors $MT_1$ and $MT_2$ are involved in synchronizing circadian rhythms and are important targets for treating sleep and mood disorders, type-2 diabetes and cancer. Here, we performed large scale structure-based virtual screening for new ligand chemotypes using recently solved high-resolution 3D crystal structures of agonist-bound MT receptors. Experimental testing of 62 screening candidates yielded the discovery of 10 new agonist chemotypes with sub-micromolar potency at MT receptors, with compound **21** reaching $EC_{50}$ of 0.36 nM. Six of these molecules displayed selectivity for $MT_2$ over $MT_1$. Moreover, two most potent agonists, including **21** and a close derivative of melatonin, **28**, had dramatically reduced arrestin recruitment at $MT_2$, while compound **37** was devoid of $G_i$ signaling at $MT_1$, implying biased signaling. This study validates the suitability of the agonist-bound orthosteric pocket in the MT receptor structures for the structure-based discovery of selective agonists.

**\*For correspondence:**
katritch@usc.edu

**Present address:** †Discovery Chemistry, Genentech Inc, South San Francisco, United States; ‡Department of Cell Biology, Neurobiology, and Anatomy, Medical College of Wisconsin, Milwaukee, United States

**Competing interests:** The authors declare that no competing interests exist.

## Introduction

The type 1A and 1B melatonin receptors ($MT_1$ and $MT_2$) are G protein-coupled receptors (GPCRs) that respond to the neurohormone melatonin (N-acetyl-5-methoxytryptamine) (*Pévet, 2016*; *Reppert et al., 1994*). Melatonin is found in all mammals, including humans, where it regulates sleep and helps to synchronize the circadian rhythm with natural light-dark cycles (*Brzezinski, 1997*; *Xie et al., 2017*). Chemically, melatonin is synthesized from serotonin in the pineal gland of the brain during darkness (*Ganguly et al., 2002*). Both $MT_1$ and $MT_2$ share canonical helical 7-transmembrane (7-TM) topology (*Johansson et al., 2019*; *Stauch et al., 2019*), although they are differentially expressed and implicated in diverse biological functions and pathologies (*Dubocovich and Markowska, 2005*). While exogenous melatonin has been commonly used for the treatment of insomnia and jetlag, more effective and long-lasting MT agonists such as ramelteon have been approved for primary chronic insomnia treatment, because of their low side-effect profile as compared to other sleeping aids such as benzodiazepines (*Hardeland et al., 2011*; *Erman et al., 2006*). Other MT agonists such as tasimelteon and agomelatine, are used for non-24-hour sleep-wake disorders in blind individuals and as an atypical anti-depressant for major depressive disorders, respectively (*Lavedan et al., 2015*; *de Bodinat et al., 2010*). Recent studies also suggest MT receptors play an essential role in learning, memory, and neuroprotection (*Liu et al., 2016*) and illustrate the potential utility of partial and selective $MT_2$ receptor agonists as antinociceptive drugs (*López-*

*Canul et al., 2015*). Moreover, $MT_2$ single nucleotide polymorphisms (SNPs) are implicated in type-2 diabetes (*Karamitri et al., 2018*) (T2D), emphasizing the importance of MT receptors in a wide variety of functions relevant to human health and the quality of life (*Karamitri and Jockers, 2019*).

Although $MT_1$ and $MT_2$ receptors have distinctive in vivo functions, most of the currently available drugs non-selectively activate both $MT_1$ and $MT_2$ receptors (*Zlotos et al., 2014*).Recent studies on melatonin receptors using partially selective $MT_2$ ligands and gene knockout approaches have shed light on difference in the biology of melatonin receptor subtypes. For example, the $MT_2$ receptor regulates non-rapid eye movement (NREM) while $MT_1$ mediates rapid eye movement (REM) phases of the vigilance state in sleep architecture (*Comai et al., 2013*; *Fisher and Sugden, 2009*; *Liu et al., 2016*). The discovery of novel and selective MT ligands may, therefore, lead to useful tool compounds for better pharmacological dissection of the melatonin system, and accelerate the development of alternatives to existing drugs (*Jockers et al., 2016*; *Zlotos, 2012*).

Recently, the three-dimensional structures of $MT_1$ and $MT_2$ were determined using an X-ray free-electron laser (XFEL), providing atomic-level details of receptor-ligand interactions (*Johansson et al., 2019*; *Stauch et al., 2019*). Although both receptors were resolved in complexes with agonists – agomelatine, 2-phenylmelatonin, 2-iodomelatonin and ramelteon, thermostabilizing mutations that were necessary for crystallization rendered these receptors functionally inactive. Therefore, the accuracy of these agonist-bound inactive structures in reproducing the active-state conformation of the orthosteric pocket, and their utility in the prospective discovery of new agonists requires further validation.

Here, we utilized the MT structural information to perform a large scale virtual ligand screen (VLS) on both $MT_1$ and $MT_2$ receptors, using libraries of 8.4 million available-for-purchase fragment-like and lead-like compounds (*Sterling and Irwin, 2015*). Subsequent experimental testing of 62 compounds selected from the top scoring molecules led to the discovery of ten new agonist chemotypes with sub-micromolar potencies, with one of them, compound **21**, displaying sub-nM agonist potency ($EC_{50}$ = 0.36 nM) in G-protein assays. Six of these hits, including the most potent one, demonstrated selectivity for $MT_2$, while five hits were partial agonists at $MT_2$. Moreover, the two most potent $MT_2$ compounds, **21** and a close derivative of melatonin, **28**, show reduced arrestin signaling, thus resulting in substantial bias towards G-protein signaling. Our results demonstrate that structure-based VLS approach can yield novel, highly potent and selective ligand chemotypes with potential utility as chemical probes with distinct properties and candidate leads for the treatment of circadian rhythm related sleep and mood disorders.

## Results

### Benchmarking receptor models

To evaluate the ability of the structure-based receptor models to recognize high-affinity melatonin receptor ligands, we performed extensive docking of a subset of known ligands of $MT_1$ and $MT_2$ receptors (*Figure 1—figure supplement 1*) into (i) the unmodified 3D structures obtained from X-ray crystallography ($MT_1$_Xtal, $MT_2$_Xtal), as well as (ii) into the receptor models where the ligand-binding pocket was optimized by conformational modeling ($MT_1$_Opt, $MT_2$_Opt). Analysis of the docking poses for the known MT ligands in both crystal structures and optimized MT receptor models showed favorable binding scores with docking poses consistent with the orientation and binding modes of crystallized ligands (*Figure 1a–d*). The major interactions include aromatic stacking of the heterocyclic core with ECL2 hydrophobic residue F179/192$^{ECL2}$ (the residue numbers for $MT_1$ and $MT_2$ listed for UniProt (*Bateman et al., 2017*) ids: P48039 and P49286, respectively, followed by superscripted Ballesteros – Weinstein numbering scheme *Ballesteros and Weinstein, 1995*), as well as hydrogen bonding interactions with N162/175$^{4.60}$ and Q181/194$^{ECL2}$ (*Johansson et al., 2019*; *Stauch et al., 2019*) . The performance of each model was then evaluated as the area under the corresponding receiver operator characteristic (ROC) curve (AUC), benchmarking the ability of these models to correctly detect ligands among decoys. The AUC values for the optimized models of MT receptors showed substantial improvement over AUC values for MT crystal structures ($MT_1$_Opt = 87 vs. $MT_1$_Xtal = 69; and $MT_2$_Opt = 82 vs. $MT_2$_Xtal = 70) (see *Figure 1e*). Overall, these results validated the improved VLS performance of the optimized models of $MT_1$ and $MT_2$ receptors, which were then used for large-scale prospective VLS.

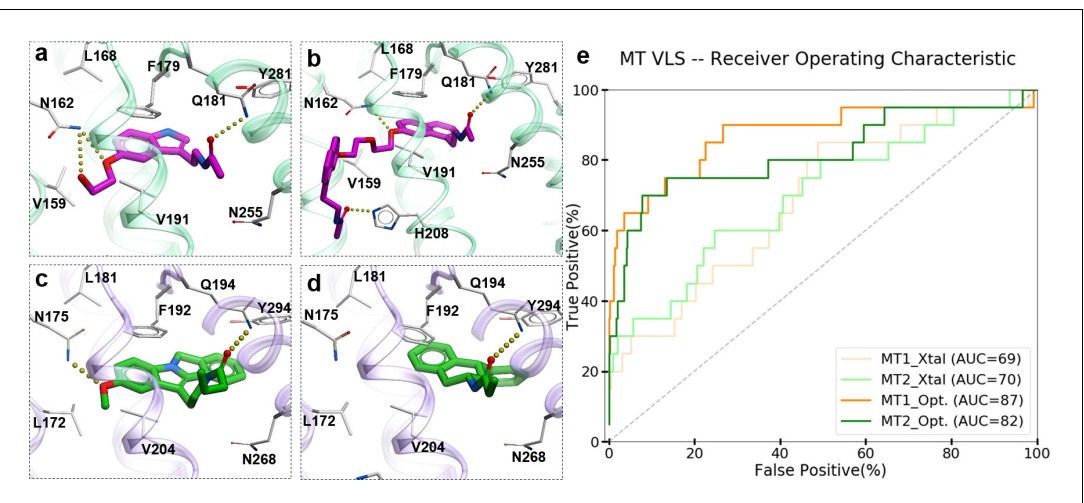

**Figure 1.** Predicted binding modes of selected known MT ligands. (**a**) 5-HEAT and (**b**) S-26131 docked into MT$_1$_Opt model; whereas (**c**) IIK-7 and (**d**) 4P-PDOT docked into MT$_2$_Opt model. (**e**) ROC plots for MT receptor crystal structures and optimized models.
The online version of this article includes the following figure supplement(s) for figure 1:

**Figure supplement 1.** Chemical structures of known (**a**) MT$_1$-selective and (**b**) MT$_2$-selective ligands, used in the benchmark.
**Figure supplement 2.** MT$_1$ crystal structure in complex with 2-PMT (PDB id: 6ME3) displaying mutated residues near orthosteric ligand binding pocket.

## Prospective VLS and candidate selection

A library of 8.4 million commercially available compounds was docked into the optimized MT$_1$_Opt and MT$_2$_Opt structural models (see Materials and methods), and for every compound, docking scores and binding interactions were predicted. The top 5000 scoring compounds were selected from these VLS runs for each receptor, which were further evaluated by redocking into both MT$_1$ and MT$_2$ receptors with increased computational sampling. The initial hit list contained 700 compounds predicted to bind to both receptors. To evaluate these top docking solutions, we created additional models of MT receptors by restoring thermostabilizing mutations (*Figure 1—figure supplement 2*) in the proximity of the orthosteric site to wild-type residues (A104$^{3.29}$G and W251$^{6.48}$F in MT$_1$; W264$^{6.48}$F in MT$_2$), and performed further conformational optimizations. We determined that the impact of these mutations on the docking results was negligible. The dock scores for selected MT ligands were better than the standard ICM VLS cutoff −32.0 kJ/mol, which is better than or comparable to the docking score of melatonin (−29.3) and other high affinity MT receptor ligands (*Johansson et al., 2019*).

To capture chemotype diversity, we selected the top 500 compounds for each receptor using chemical clustering in combination with docking scores. A final set of 62 compounds (23 from only MT$_1$ VLS; 25 from only MT$_2$ VLS; 14 from both MT$_1$ and MT$_2$ VLS) were selected for purchase based on a multidimensional composite criterion accounting for compound novelty, chemical diversity, well-defined interaction patterns with binding site residues N162/175$^{4.60}$ and/or Q181/194$^{ECL2}$, and interaction similarities with ligands observed in the crystal structures (See *Figure 2*; *Supplementary file 1* Table S1).

Most of the compounds represented new chemotypes with Tanimoto chemical distance values >0.22 (*Abagyan et al., 2016*), separating them from known high-affinity MT ligands available in CHEMBL24 (*Gaulton et al., 2017*). We also chose a close analog of melatonin – compound **28** (Tanimoto distance = 0.05), which to our knowledge, has not yet been characterized as a ligand for MT receptors (*Gaulton et al., 2017*; *Kim et al., 2019*). Compound **28** served as an additional positive control, which also helped us to evaluate the effect of a single chemical group substitution in melatonin on the binding and function at the MT receptors.

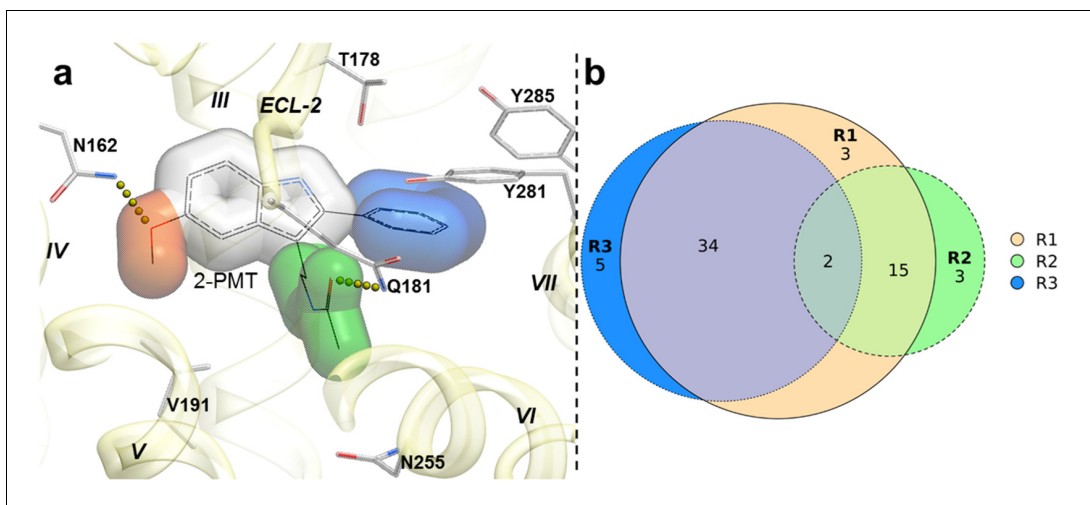

**Figure 2.** Structural features in selected hit candidate compounds. (**a**) 2-phenylmelatonin in complex with MT1 receptor with the topology of chemical features shown as colored spheres indicating R1 (orange) = 5-methoxy, R2 (green) = alkylamido chain, and R3 (blue) = 2-phenyl substitutions, (**b**) Venn diagram summarizing the topologically equivalent chemical features in selected 62 candidate compounds from $MT_1$ and $MT_2$ VLS.

The online version of this article includes the following figure supplement(s) for figure 2:

**Figure supplement 1.** Histogram of predicted logP values of known high-affinity MT ligands ($N_{ref}$ = 515) from ChEMBL and selected MT ligands ($N_{sel}$ = 30).

## Experimental hit identification and validation

The selected 62 candidate compounds from VLS were tested experimentally for binding and functional profiles in both MT receptors. Eleven compounds (***Table 1***; ***Figure 3***) demonstrated sub-

**Table 1.** Hit compounds from VLS with $G_{i/o}$ mediated potency $EC_{50}$ <1 µM for at least one MT receptor.

| Compound | $pK_i \pm$ SEM* | $pEC_{50} \pm$ SEM | $EC_{50}$(nM) | $E_{max}^{\dagger} \pm$ SEM | $LE^{\ddagger c}$ | $pK_i \pm$ SEM | $pEC_{50} \pm$ SEM | $EC_{50}$(nM) | $E_{max} \pm$ SEM | LE | Selectivity$^{\S}$ | Tanimoto$^{\P}$ |
|---|---|---|---|---|---|---|---|---|---|---|---|---|
| | $MT_1$ | | | | | $MT_2$ | | | | | $MT_2/MT_1$ | |
| 21 | 6.31 ± 0.11 | 7.91 ± 0.05 | **12.0** | 93.8 ± 2.5 | 0.69 | 6.91 ± 0.05 | 9.44 ± 0.08 | **0.36** | 86.1 ± 3.2 | 0.83 | 30.6 | 0.50 |
| 23 | 5.42 ± 0.03 | 7.16 ± 0.09 | **57.5** | 96.9 ± 5.3 | 0.56 | 5.56 ± 0.13 | 7.69 ± 0.08 | **20.42** | 91.7 ± 3.0 | 0.60 | 2.7 | 0.22 |
| 28 | 7.78 ± 0.10 | 10.39 ± 0.04 | **0.04** | 95.3 ± 2.6 | 0.86 | 7.63 ± 0.08 | 10.35 ± 0.10 | **0.04** | *69.4 ± 4.0* | 0.85 | 0.7 | 0.05 |
| 29 | 5.22 ± 0.07 | 6.83 ± 0.06 | 144.5 | 87.5 ± 4.5 | 0.53 | 5.61 ± 0.05 | 7.46 ± 0.10 | **34.67** | *69.4 ± 8.0* | 0.58 | 3.3 | 0.43 |
| 37 | 5.07 ± 0.13 | ND | >30000 | ND | ND | 5.45 ± 0.10 | 6.85 ± 0.19 | 141.25 | *61.1 ± 9.1* | 0.53 | >1000.0 | 0.57 |
| 44 | 4.19 ± 0.36 | 3.33 ± 0.36 | 57544.0 | 72.8 ± 4.7 | 0.33 | 4.95 ± 0.30 | 6.58 ± 0.13 | 263.03 | 88.9 ± 6.3 | 0.51 | 267.2 | 0.59 |
| 45 | 4.54 ± 0.15 | 5.06 ± 0.12 | 8709.6 | 90.6 ± 14.3 | 0.44 | 5.26 ± 0.19 | 6.37 ± 0.13 | 426.58 | 75.0 ± 7.4 | 0.56 | 16.9 | 0.59 |
| 47 | 4.58 ± 0.07 | 5.25 ± 0.16 | 2344.2 | 112.4 ± 5.2 | 0.46 | 5.91 ± 0.12 | 7.99 ± 0.10 | **10.23** | 91.7 ± 3.0 | 0.66 | 186.9 | 0.60 |
| 54 | 5.03 ± 0.06 | 6.06 ± 0.07 | 741.3 | 82.8 ± 4.3 | 0.54 | 5.56 ± 0.10 | 7.74 ± 0.10 | **18.20** | 75.0 ± 3.7 | 0.68 | 36.9 | 0.43 |
| 57 | 4.84 ± 0.03 | 5.72 ± 0.11 | 1778.3 | 87.5 ± 9.1 | 0.47 | 5.37 ± 0.04 | 6.88 ± 0.15 | 131.83 | *66.7 ± 8.3* | 0.57 | 10.3 | 0.53 |
| 62 | 4.32 ± 0.11 | 4.39 ± 0.42 | 42658.0 | 54.1 ± 10.0 | 0.36 | 5.49 ± 0.33 | 7.28 ± 0.14 | **52.48** | *58.3 ± 4.8* | 0.60 | 875.9 | 0.64 |
| Melatonin | 9.06 ± 0.14 | 11.38 ± 0.06 | **0.004** | 100.0 ± 5.6 | 0.93 | 9.27 ± 0.14 | 10.30 ± 0.14 | **0.05** | 100.0 ± 5.6 | 0.84 | 0.1 | 0.00 |

Standard error of the mean, N = 3.

$^{\dagger}$ Activation compared to melatonin.

Ligand efficiency (based on $EC_{50}$).

$^{\S}$Selectivity in folds (calculated as: Antilog (log($E_{max}/EC_{50}$) $MT_2$- log ($E_{max}/EC_{50}$) $MT_1$)). $MT_1$ selectivity is shown as underlined values.

$^{\P}$ Tanimoto distance from closest MT receptor ligands in ChEMBL database with pAct >6. Hits with $EC_{50}$ <100 nM are displayed in **bold**, and with $E_{max}$ <70% in *italic*.

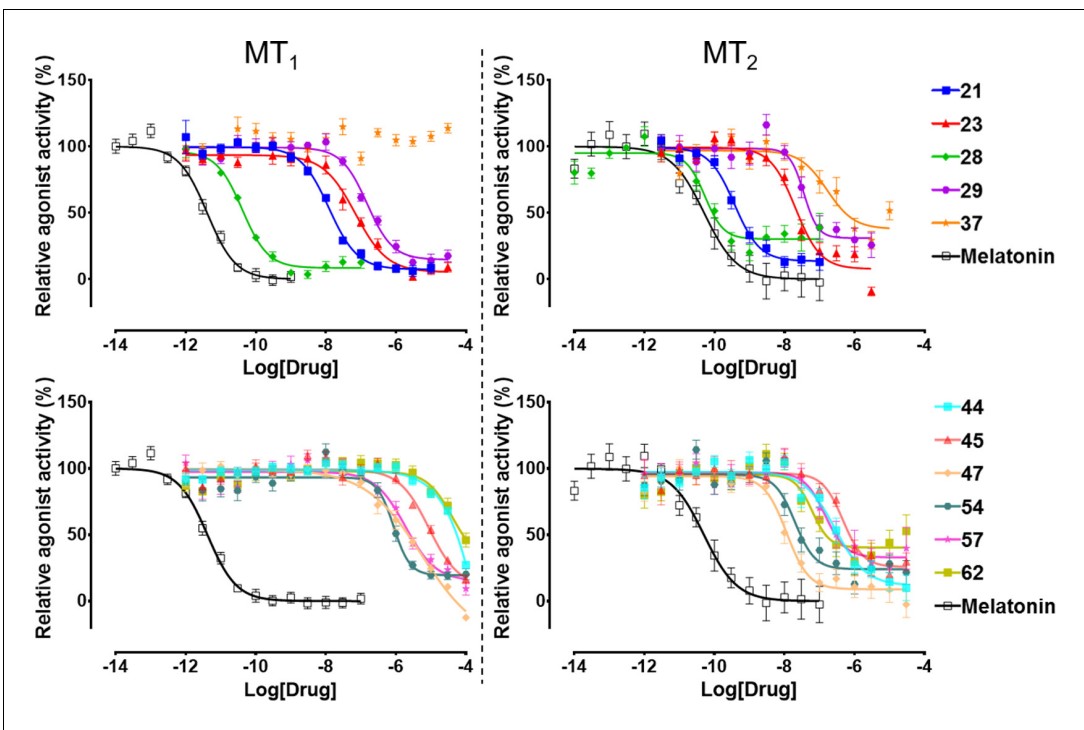

**Figure 3.** Functional characterization of selected VLS hits for agonist activity at $MT_1$ and $MT_2$ receptors in $G_{i/o}$-mediated cAMP production inhibition assays. Results were normalized to the $E_{max}$ value (%) of receptor activation by melatonin. These VLS hits showed no activity at control HEK293 T cells without transiently transfected $MT_1$ or $MT_2$ receptors (results not shown).

The online version of this article includes the following figure supplement(s) for figure 3:

**Figure supplement 1.** Radioligand $^3$H-Melatonin competition binding data for selected hit compounds.
**Figure supplement 2.** Tango assay measuring agonist-induced β-arrestin recruitment by $MT_1$ receptor.
**Figure supplement 3.** Tango assay measuring agonist-induced β-arrestin recruitment by $MT_2$ receptor.

micromolar potencies in $G_{i/o}$ mediated signaling assays (18% hit rate). Ten of these eleven compounds also showed binding affinities $K_i$ <10 µM in a competition binding assay (*Figure 3—figure supplement 1*). The melatonin derivative **28** identified by VLS was as potent as melatonin itself in $MT_2$ ($EC_{50}$ = 0.04 nM) and had the same potency ($EC_{50}$ = 0.04 nM) at $MT_1$. The most potent new chemotype, **21,** displayed an $EC_{50}$ = 0.36 nM for $MT_2$, with a 30-fold selectivity over $MT_1$ receptor ($MT_1 EC_{50}$ = 12 nM). Overall, seven hits had $EC_{50}$ <100 nM for at least one of the melatonin receptors. Similar to other low molecular weight MT ligands, most of the hits identified belong to a library of fragment-like compounds with molecular weights less than 250 Da, and have exceptionally high ligand efficiency (LE), far exceeding the ~0.3 value considered as a standard for a promising lead. For example, compound **21** (Mol. Wt. = 224 Da) had the highest LE values of 0.83 kcal/mol per non-hydrogen atom for $MT_2$ and 0.69 kcal/mol per non-hydrogen atom for $MT_1$ receptors (*Hopkins et al., 2004*; *Hopkins et al., 2014*). The excellent LE of these molecules allows the potential for further optimization of their drug-like properties.

## Chemical and conformational diversity of hits

Most of the hit compounds, as shown in *Chemical structure 1*, are novel and display diverse chemotypes distinct from known high-affinity MT ligands (ChEMBL, pAct >6.0), with Tanimoto distance exceeding 0.4 for all but two ligands (**28** and **23**). While the majority of known MT agonists reported in ChEMBL have either substituted indene or naphthalene core, only two of the eleven hits reported here have fused heterocycles and several others have two substituted aromatic rings connected by a flexible chain. Most compounds have diverse substitutions at positions topologically equivalent to the 5-methoxy, acetylamido and C2 position of melatonin (*Figure 2*). Two compounds– **21** and **29**–

**Chemical structure 1.** Chemical structures of hit compounds with EC$_{50}$ <1 µM at the MT receptors.

have a pyrimidine scaffold, whereas four compounds– **23**, **37**, **44**, and **57**– have a methoxyphenyl group in place of the 5-methoxy indole scaffold in melatonin. Another interesting core is the cyclopentyl-fused thienopyridine of compound **45**. Only 2 compounds, **28** and **54**, have substituted indoles similar to melatonin.

The predicted binding poses of the selected hit compounds in their docking models of MT receptors are shown in *Figure 4*. Nine out of eleven hits have methoxy or a similar group predicted to make hydrogen bonding interactions with N162/175$^{4.60}$, which was found to be a critical residue for receptor activation, despite playing a limited role in ligand affinity or structural stability of the receptor (*Stauch et al., 2019*).

Furthermore, seven of the hits were predicted to form hydrogen bonding interactions with Q181/194$^{ECL2}$ similar to alkylamide tail of melatonin. Five hits were predicted to occupy a significant space in the pocket flanked by TMs-II, III, and VII forming hydrophobic interactions, especially with residues Y281/294$^{7.38}$ and Y285/298$^{7.43}$, as previously found in the MT receptor structures (*Johansson et al., 2019*; *Stauch et al., 2019*). These hydrophobic interactions are similar to those formed by the phenyl moiety of 2-phenylmelatonin. Both types of hydrogen bonding and hydrophobic interactions were found to be critical for a ligand's steric fit into the MT receptor binding pocket and are the primary determinants of ligand affinity.

## Structural basis of subtype selectivity of the hits

Six of the identified hits were found to be at least 30 fold more potent at MT$_2$ compared to MT$_1$ in the G$_{i/o}$-mediated cAMP inhibition assays. Among the hits reported here with novel scaffolds, compound **21** has the highest potency for both MT$_2$ (EC$_{50}$ = 0.36 nM) and MT$_1$ (EC$_{50}$ = 12 nM). Compound **21** was predicted to bind both the MT receptors in a similar orientation by forming hydrogen bonding interactions with N162/175$^{4.60}$ and Q181/194$^{ECL2}$ with its methoxy anchor and acetylamido tail, respectively. These interactions had been reported to be critical for ligand affinity and potency at the MT receptors (*Johansson et al., 2019*; *Stauch et al., 2019*).

Other compounds also possess remarkable MT$_2$ selectivity. For example, compound **47** is 187-fold selective towards MT$_2$ (EC$_{50}$ = 10 nM for MT$_2$, and 2.34 µM for MT$_1$, respectively). The pyrrole ring mimics the indole ring of melatonin, the amide group forms hydrogen bonding with N162/

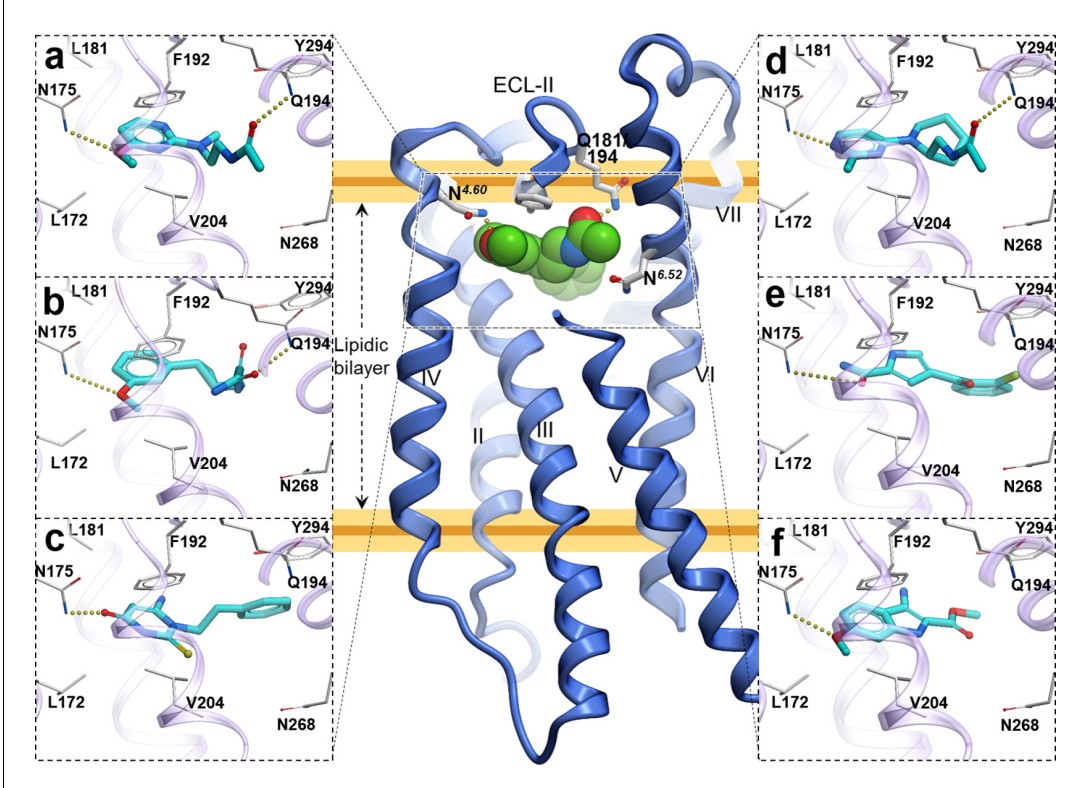

**Figure 4.** Predicted binding poses for top six new chemotypes discovered with VLS. (a) **21**, (b) **23**, (c) **62**, (d) **29**, (e) **47** and (f) **54** in the MT$_2$ receptor (purple). The center panel shows a canonical 7-TM receptor structure of MT$_2$ receptor (blue helices; part of TM-V is not displayed for clarity) in complex with 2-phenylmelatonin shown as green spheres (PDB id: 6ME6).

175$^{4.60}$ and the chlorophenyl group forms hydrophobic interactions with ECL2 and TM-II, III, and VII residues (*Figure 4*). Despite the lack of polar interactions with Q181/194$^{ECL2}$, the compound displays sub-micromolar potency for MT$_2$. Similarly, compound **45** also lacks a substitution topologically equivalent to acetylamido tail of melatonin (R2 feature) and yet has a sub-micromolar potency and 17-fold selectivity for MT$_2$ (EC$_{50}$ = 427 nM). In contrast, compound **44** was predicted to form interactions with Q181/194$^{ECL2}$, but it lacks an R3 equivalent substitution, which still makes it 267-fold selective for MT$_2$ (EC$_{50}$ = 263 nM). These findings suggest that either R2 or R3 could be sufficient in maintaining the potency and selectivity at MT$_2$.

## Functional selectivity of the hit ligands

All the discovered hits show activity as agonists in G$_{i/o}$-protein signaling assays at both MT$_1$ and MT$_2$ receptors. At the same time, some compounds show functional profiles notably distinct from full and balanced agonism, especially at MT$_2$. Thus, four of the hits, **28**, **29**, **57**, and **62** had their efficacy (E$_{max}$) reduced to less than 70% in MT$_2$, and are therefore considered partial agonists (*Audinot et al., 2003*). The identified hits were also evaluated for their β-arrestin recruitment (*Figure 3—figure supplements 2* and *3*), with the comparative analysis of G-protein and β-arrestin activity shown in *Figure 5*. In the case of the MT$_1$ receptor, there are no significant deviations from the overall balanced G-protein/Arrestin signaling profiles for most compounds (*Figure 5a*). One exception is compound **37**, which completely lacks G-protein signaling, though it still binds to MT$_1$ and displays substantial β-arrestin activity. In the case of MT$_2$, however, there are several compounds that show a marked reduction in β-arrestin signaling compared to G-protein, especially compounds **21** and **28**, which show bias factors of 15.5 and 33.9, respectively (*Figure 5b*). These results suggest that MT ligands may show rather distinct functional bias profiles in G-protein and β-arrestin

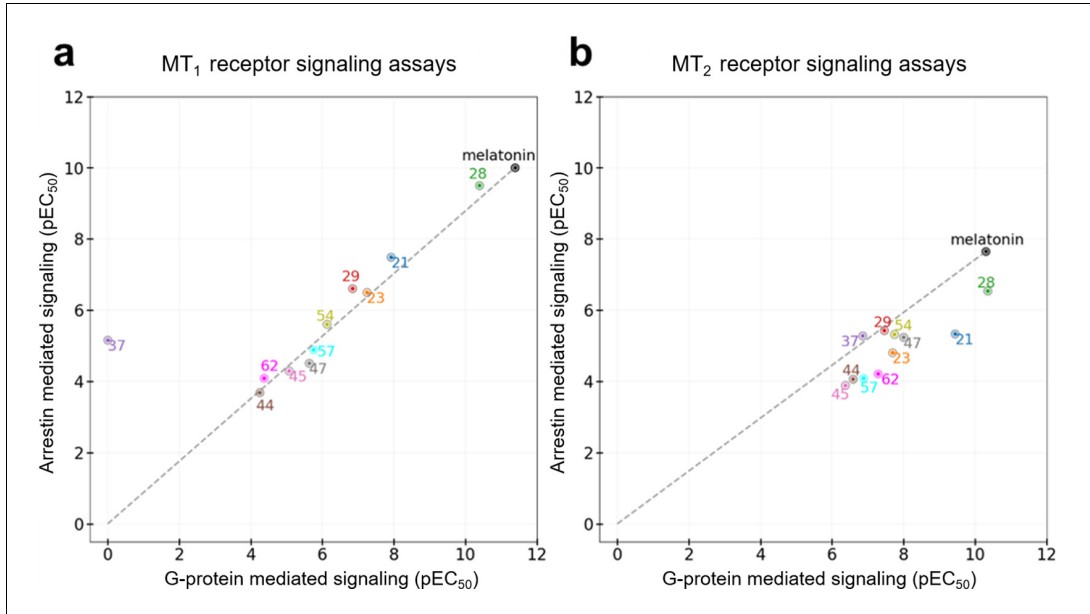

**Figure 5.** Functional selectivity of selected VLS hits at $MT_1$ (**a**) and $MT_2$ (**b**) receptors. The $pEC_{50}$ values of the ligands in G-Protein and Arrestin-mediated signaling assays are shown. The dashed lines for each receptor trace the melatonin datapoint to the origin, with compounds far above or below the line showing functional selectivity.

signaling, as observed in many other GPCRs (*Kenakin, 2019*; *Roth, 2019*), though the biological importance of this bias in MT remains to be investigated (*Cecon et al., 2018*).

To gain more insights into these variations in ligand activity at MT receptors, we analyzed conformational differences among ligand-receptor pairs for these compounds. As the hit compounds are fragment–like and may attain multiple energetically-favorable poses at the orthosteric site upon docking, the specific conformational features driving partial agonism remain unclear. However, analysis of compounds **28** and **37** with the most pronounced bias to G-protein in $MT_2$ and β-arrestin in $MT_1$, respectively, suggests possible explanations for these phenomena.

Compound **28** is very similar to melatonin, except the amide is replaced by a urea. This substitution renders compound **28** as a partial agonist at $MT_2$ ($E_{max}$ = 69.4%) while largely maintaining full agonism at $MT_1$ ($E_{max}$ = 95.3%). Docking predictions suggest that compound **28** assumes an orientation in the binding pocket similar to 2-phenylmelatonin with subtle differences, as shown in *Figure 6*. In $MT_2$ orthosteric site, the urea of compound **28** forms hydrogen bonding interactions with the side chains of polar residues $Q194^{ECL2}$ and $N268^{6.52}$ owing to its additional nitrogen. Such interactions, however, are energetically unfavorable in the case of $MT_1$ with possible steric clashes (*Johansson et al., 2019*; *Stauch et al., 2019*). Instead, the interactions of acetylamido group of melatonin with $Q181^{ECL2}$ are replaced by a hydrogen bond between $Y281^{7.38}$ and oxygen from the urea in compound **28**. These interactions become favorable in $MT_1$ as the residue $Y281^{7.38}$ is rotated towards TM-VI placing it 4 Å away from $T178^{ECL2}$. In the case of $MT_2$, however, residues $Y294^{7.38}$ – $T191^{ECL2}$ are 3 Å apart forming an intermolecular hydrogen bond with $Y281^{7.38}$ oriented away from TM-VI allowing favorable orientation of $Q194^{ECL2}$ to form a hydrogen bond with compound **28**.

A similar pattern of ligand-receptor interaction is observed from the docking of the most selective compound **37** into MT receptors. Compound **37** has a distinct and much bulkier substitution with a 3-cyclopropyl-1,2,4-oxadiazol group (R2 feature). In $MT_1$, this oxadiazol group is predicted to form hydrogen bonds with $Q181^{ECL2}$, $T178^{ECL2}$ and $Y281^{7.38}$, while the cyclopropyl group is predicted to fit in the sub pocket formed by ECL2, TM-V, and VII. These interaction pattern changes in $MT_2$, where the prominent hydrogen bonding between side chains of $T191^{ECL2}$ and $Y294^{7.38}$ is formed, precluding hydrogen bonding of these residues to compound **37**. Moreover, the methoxy group (R1 feature) of the compound, which maintains a hydrogen bond with $N175^{4.60}$ in $MT_2$ is lost with $N162^{4.60}$ side chain in MT1, due to a subtle shift of the compound. Indeed, the methoxy – $N162^{4.60}$

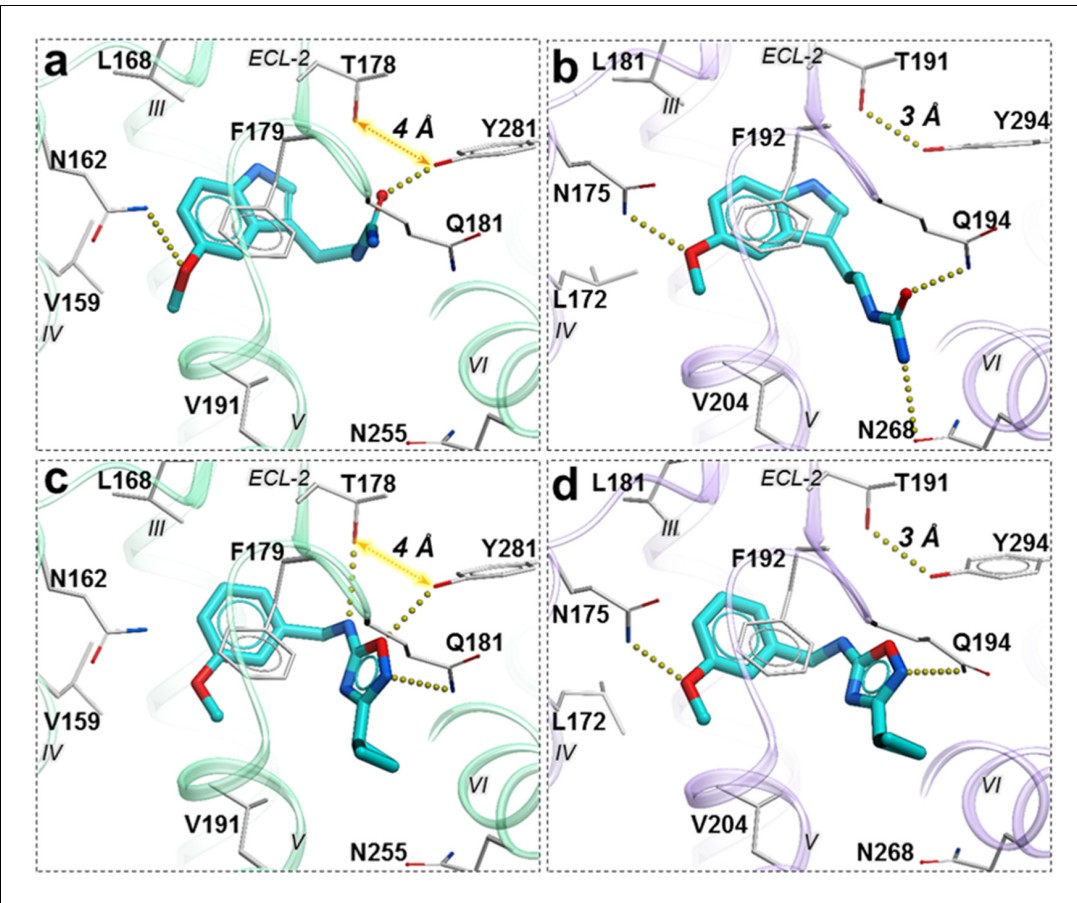

**Figure 6.** Predicted binding poses for compounds **28** (a, b), and **37** (c, d) in $MT_1$ (light green) and $MT_2$ (lavender) receptors, respectively. The red dotted lines with arrows indicate a missing hydrogen bond between residues T178 and Y281 in $MT_1$ receptor, while the yellow dots show hydrogen bonding interactions.

The online version of this article includes the following figure supplement(s) for figure 6:

**Figure supplement 1.** Tango functional assays with MT1 mutants for melatonin.

**Figure supplement 2.** Binding activities of Compounds 21, 28 and 37 at a set of 47 potential off-targets.

interaction is found to be critical in receptor activation (*Stauch et al., 2019*), and loss of this interaction is likely to explain lack of activity of compound **37** in $G_i$-mediated signaling at $MT_1$. This interaction difference, however, does not seem to affect the β-arrestin mediated signaling by compound **37** in $MT_1$. This peculiar feature of **37** is supported by our mutational studies, where N162Q mutation in $MT_1$ actually increased potency of **37** slightly (2-fold), while Y281F, as expected, reduced potency by over 10x (*Figure 6—figure supplement 1*). For comparison, **28** drastically (>100 fold) reduced potency in both $MT_1$ mutants N162Q and Y281F. Taken together, these results support a key role of N162/175[4.60] anchoring interactions in $G_i$-mediated receptor activation, and also suggest a distinct role of residues Y281/294[7.38] in governing ligand bias at MT receptors. Further analysis, including mutation and SAR studies of compound **37** derivatives are needed for comprehensive validation of this hypothetical mechanism in future studies.

## Off-target profiling

To verify the ligand selectivity, the lead compounds **21**, **28**, and **37**, were subjected to binding profiling at a panel of 47 common drug targets (including many GPCRs and neurotransmitter transporters). At a final concentration of 10 µM, they did not show any substantial binding at these targets, except for compound **28** that displayed over 50% inhibition at three 5-HT receptors (*Figure 6—figure supplement 2*), with binding affinity of 851 nM ($pK_i$ = 6.07 ± 0.11) for 5-HT$_{1A}$, 1525 nM

($pK_i$ = 5.82 ± 0.02) for 5-HT$_{1D}$, and 286 nM ($pK_i$ = 6.54 ± 0.03) for 5-HT$_{7A}$. Assessment of functional activity of **28** in G$_s$-mediated cAMP production shows that it is a weak partial agonist with EC$_{50}$ = 1819 nM, as compared to 0.04 nM at both MT$_1$ and MT$_2$ receptors. These results demonstrated that the lead compounds **21**, **28**, and **37** act specifically at MT receptors and show high selectivity for MT$_1$ or MT$_2$ over many common drug targets.

## Discussion

The discovery of potent and selective MT ligands with novel chemotypes holds promise for the development of next-generation drugs to treat circadian rhythm and mood disorders, pain, insomnia, type-2 diabetes, and cancer (*Karamitri and Jockers, 2019*; *Liu et al., 2016*). Herein, we utilized the recently solved 3D structures of the melatonin receptors, in complex with the agonist 2-phenyl-melatonin, (*Johansson et al., 2019*; *Stauch et al., 2019*) to perform prospective virtual ligand screening of large fragment-like compound libraries. This approach resulted in the discovery of ten new chemotypes of potent agonists, both full and partial, for MT$_1$ and MT$_2$. The number of sub-micromolar hits and potency of the best among them is one of the highest reported for a VLS campaign in class A GPCRs (*Lyu et al., 2019*) and in-line with another VLS screen for MT receptors, published while this study was in revision (*Stein et al., 2020*). This is remarkable, considering that most GPCR structures have a limited capacity to distinguish agonists vs. antagonists (*Costanzi and Vilar, 2012*; *Weiss et al., 2018*) and prospective VLS campaigns often result in antagonists even when an agonist-bound VLS model is used (*Lyu et al., 2019*; *Roth, 2019*).

There are several factors, related to both the VLS procedure and the intrinsic properties of the MT receptors, that likely contributed to the high hit rate and agonistic potency of the hits in our study. Thus, the high quality of the initial crystal structure, further improved by ligand-guided optimization of the pocket for VLS, has been critical for the success of our previous VLS campaigns, and likely played a similar role here (*Katritch et al., 2010*; *Lane et al., 2013*; *Zheng et al., 2017*). At the same time, some intrinsic properties of MT receptors also likely facilitated successful VLS for agonists. As we mentioned above, endogenous ligand melatonin itself has unusually high picomolar potency at MT receptors (~4 pM at MT$_1$ and 50 pM at MT$_2$, see *Table 1*). Melatonin and most other high-potency ligands are small (<250 DA) and yet they still fully occupy the very small, enclosed MT pocket. Chemical space of such size-limited fragment-like libraries is much smaller than the usual drug-like space, and can be more exhaustively searched, likely resulting in higher hit rates. Moreover, most known MT receptor ligands show agonist activity, while antagonists of similar potency are notoriously hard to find, suggesting that agonists may be intrinsically preferred ligands for MT receptors (*Jockers et al., 2016*).

Two of the hit compounds, **37** and **62**, are MT$_2$-selective partial agonists, which may have a desirable profile for eliciting antinociceptive effects mediated by melatonin receptors, and may be potentially useful in developing novel analgesics for pain management with reduced side effects (*López-Canul et al., 2015*). Of note, while some of the newly discovered hits are selective for MT$_2$, none of the hits in this study had substantial MT$_1$ selectivity. This may be explained by the lack of a bulky chemical group at the R1 position, which are known to confer strong MT$_1$ selectivity, e.g. in bitopic CTL 01–05-B-A05 ligand that stretches out of the pocket via narrow side channel (*Stauch et al., 2019*). Design of such bitopic ligands with MT$_1$ selective chemotypes would need to explore larger compounds (MW >500), which were not considered in the current VLS screen.

This study represents a successful application of structure-based VLS to identify agonists with novel chemotypes, sub-nanomolar potencies, and a high degree of receptor subtype selectivity for a class A GPCR (*Lyu et al., 2019*; *Wang et al., 2017*). This study also represents a successful implementation of molecular modeling and structure-based virtual screening, aimed at the melatonin receptors, facilitated by the availability of high-quality structures capturing vital ligand-receptor interactions (*Alkozi et al., 2018*; *Johansson et al., 2019*; *Stauch et al., 2019*). Prevalence of agonists in the hit set suggests the importance of activated, agonist-bound conformations of the orthosteric pocket models for successful agonist screening. Note that, even though the receptors were thermostabilized by 9 and 8 point mutations (MT$_1$/MT$_2$, respectively) to aid crystallization rendering the receptor conformations inactive on the intracellular side, the agonist-bound orthosteric pockets remain relevant for structure-based drug discovery applications. Our benchmarking also corroborated the important role of ligand guided receptor optimization (LiBERO) (*Katritch et al., 2012*) in

improving the outcomes of a structure-based VLS, similar to some of our previous VLS campaigns (*Lane et al., 2013*; *Zheng et al., 2017*). Another critical aspect of this successful VLS is the discovery of novel chemotypes with reliable docking poses. With our screening library assembled to be fragment-like with regards to molecular weights, our hits are diverse and amenable to chemical optimization to improve their pharmacological profiles. Thus, our results also illustrate the utility of fragment-like compounds in the early stages of drug discovery.

The chemical diversity, selectivity, high potency and agonist activities of the identified hits serve as a valuable starting point for the development of tool compounds to explore the biology of melatonin receptors. With the potential for selective modulation over the melatonin receptor subtype-mediated biology, these novel chemotypes could provide new leads for the development of next-generation treatments for insomnia, pain, sleep and mood-related disorders, type 2 diabetes, and cancer.

## Materials and methods

**Key resources table**

| Reagent type (species) or resource | Designation | Source or reference | Identifiers | Additional information |
|---|---|---|---|---|
| Cell line (*Homo sapiens*) | HTLA cells (HEKT based) | PMID:25895059 | | |
| Transfected construct (*Homo sapiens*) | MTNR1A | PMID:25895059 | AddGene #66443 | |
| Recombinant DNA reagent, PCR primers | MTNR1A N162Q Forward | This paper | | CTGCCGTCCTGCCGcaaCTGAGGGCAGGCAC |
| Recombinant DNA reagent, PCR primers | MTNR1A N162Q Reverse | This paper | | GTGCCTGCCCTCAGttgCGGCAGGACGGCAG |
| Recombinant DNA reagent, PCR primers | MTNR1A Y281F Forward | This paper | | GTTCGTAGCGAGCTtCTACATGGCTTAC |
| Recombinant DNA reagent, PCR primers | MTNR1A Y281F Reverse | This paper | | GTAAGCCATGTAGaAGCTCGCTACGAAC |
| Commercial assay or kit | BrightGlo Reagent | Promega.com | Cat # E2610 | |
| Chemical compound, drug | Hit compounds | Enamine, Molport, Chembridge | | See 62 compounds listed in *Supplementary file 1* |
| Software, algorithm | ICM-Pro, V3.8–7 | Molsof.com | | |
| cell line (*Homo sapiens*) | HEK293 T | ATCC | CRL-11268 | |
| transfected construct (*Homo sapiens*) | Human MT1 | PMID:31019306 | | |
| Chemical compound, drug | Luciferin | Goldbio.com | Cat#: LUCNA-1G | |

## Receptor model preparation and optimization

X-ray crystal structures of MT$_1$ (*Stauch et al., 2019*) and MT$_2$ (*Johansson et al., 2019*) receptors in complex with 2-phenylmelatonin (PDB IDs *Berman et al., 2000*: 6ME3, 6ME6) were used to generate virtual screening models. Both structures were converted from PDB coordinates to ICM objects using the object conversion protocol implemented in ICM-Pro (*Abagyan et al., 2016*). This process

includes the addition of hydrogens and assignments of secondary structures, the energetically favorable protonation states to His, Asn and Gln side chains, and of formal charges to the ligand in a complex with the receptor, followed by local minimization of polar hydrogens using energy minimization protocols in ICM-Pro. The orthosteric ligand-binding pocket was further optimized with energy-based global optimization in ICM using Biased Probability Monte-Carlo (BPMC), where the orthosteric ligand and amino acid side chains within 5 Å radius were kept flexible and co-optimized (*Abagyan and Totrov, 1994*), as described in LiBERO protocol (*Katritch et al., 2012*) and its previous applications (*Lane et al., 2013*; *Zheng et al., 2017*). To validate the models, a set of 20 known MT receptor ligands were selected from ChEMBL database (*Gaulton et al., 2017*) along with 780 MT receptor decoys selected for each $MT_1$ and $MT_2$ receptor from GPCR decoy database (GDD) (*Gatica and Cavasotto, 2012*) and docked into crystal structures, and optimized ligand models of MT receptors. Following the previously described ligand guided receptor binding pocket optimization protocol, the Receiver Operator Characteristic curves (ROC) were plotted based on the True Positive Rates (TPR) and False Positive Rates (FPR) (*Katritch et al., 2012*) to evaluate the model optimization. The AUC values were calculated as the areas under these ROC curves and used as a model selection criteria for prospective VLS runs. The RMSD values of ligand binding pocket side chain heavy atoms for $MT_1$ and $MT_2$ were 0.51 Å and 0.76 Å, respectively, compared to their corresponding crystal structures.

To perform additional evaluation of screening results with the thermostabilizing mutants in the proximity of the orthosteric site, as displayed in *Figure 1—figure supplement 2*, were restored to wild-type (WT) residues. The Phe residue at F251/264[6.48] located 4.3 Å from the ligand was mutated to Trp, followed by local minimization of side-chain conformations using energy-based sampling and minimization protocols (*Abagyan and Totrov, 1994*). Similarly, A104[3.29], located 5.2 Å was also restored to Gly in the $MT_1$ receptor model. Docking to this model suggests that these stabilizing mutations do not substantially impact the binding of known ligands and selected hit candidates into the orthosteric pocket.

## Screening library

We selected a subset of commercially available (in-stock and on-demand) fragment-like compounds from the ZINC database with physicochemical properties similar to already reported melatonin receptor ligands (*Gaulton et al., 2017*; *Sterling and Irwin, 2015*). We considered compounds with molecular weight ≤250 Da and logP values ranging 1 to 5 based on the logP data of high-affinity MT ligands (*Figure 2—figure supplement 1*). The initial dataset comprised of ~10 million compounds was converted from SMILES to 3D format, and formal charges were assigned. This set was further reduced to 8.4 million compounds after applying additional filters for net charges (between −1 to 1) and removing compounds with highly reactive functional groups and promiscuous PAINS chemotypes ('molPAINS' and 'bad groups' in ICM-Pro v.3.8–6) (*Baell and Holloway, 2010*).

## Virtual ligand screening

The VLS of 8.4 million compounds library for $MT_1$ and $MT_2$ models were performed using the VLS protocol in ICM-Pro (*Abagyan et al., 1994*). The receptor energy potential maps were calculated using a fine potential grid (0.5 Å). Several energy terms, including van der Waals, hydrophobic, electrostatic and hydrogen bonding interactions were considered for map calculations. Full torsional flexibility of ligands was allowed, and their internal conformational strain was considered while the receptor atoms were assigned rigid for docking. The docking was performed using BPMC conformational sampling and energy minimization protocol implemented in ICM-Pro for scoring and finding the best docking solutions at the default effort level 1. These top compounds were further docked into corresponding MT receptor models with an increased sampling effort value of 3. Each VLS run for the 8.4 million compound library used 32,000 CPU core hours on 3 Linux workstations with a total of 192 CPU cores. The chemical similarity of selected compounds was calculated using Tanimoto chemical distance function 'Distance(chem1 chem2)', available in Molsoft's ICM-Pro (*Totrov, 2008*). The fingerprints in this function use a combination of ECFP and linear fingerprints as described in ICM-Pro manual (http://www.molsoft.com/icm/fingerprints.html).

## Binding and functional assays

### Radioligand binding assays

All compounds for in vitro testing were purchased from Enamine, Molport, and Chembridge in stock libraries, with verified identity and guaranteed purity of >95% (37 compounds) or >90% (25 compounds), see *Supplementary file 2* for compound QC data).

The Radioligand binding assays were conducted by the NIMH Psychoactive Drug Screening Program (PDSP). The NIMH PDSP is directed by Bryan L. Roth, MD, PhD, at the University of North Carolina at Chapel Hill, North Carolina, and Program Officer Jamie Driscoll at NIMH, Bethesda, MD. Binding assay procedures and protocols are also available online at http://pdspdb.unc.edu/pdspWeb/?site=assays. All the radioligand binding assays were performed using membranes prepared from transiently transfected HEK293T cells (purchased from ATCC, CRL-11268, authenticated by the supplier using morphology, growth characteristics, and STR profiling and certified mycoplasma-free) and in standard binding buffer (50 mM Tris, 10 mM $MgCl_2$, 0.1 mM EDTA, 0.1% BSA, 0.01% ascorbic acid, pH 7.4) as recently reported (*Stauch et al., 2019*). [$^3$H]melatonin (PerkinElmer, specific activity = 77.4–84.7 Ci/mmol) is used as the radioligand. Competitive binding assays were performed with various concentrations of test compounds (100 fM to 10 µM), [$^3$H]melatonin (0.2–1.7 nM), and $MT_1$ or $MT_2$ membranes in a total volume of 150 µL. Assay plates were sealed and incubated for 4 hr at 37°C in a humidified incubator until harvesting. Plates were harvested using vacuum filtration onto 0.3% polyethyleneimine pre-soaked 96-well Filtermat A (PerkinElmer) and washed three times with cold wash buffer (50 mM Tris, pH 7.4). Filters were dried and melted with a scintillation cocktail (Meltilex, PerkinElmer). Radioactivity was counted using a Wallac TriLux Microbeta counter (PerkinElmer). Results were analyzed using GraphPad Prism 7.0.

## Signaling assays

### $G_s$ and $G_{i/o}$-cAMP assays

GloSensor cAMP assays were conducted according to the recently published procedure (*Stauch et al., 2019*) with minor modifications. Briefly, HEK293 T cells (as above) were transiently co-transfected with receptor ($MT_1$ or $MT_2$) and GloSensor cAMP (Promega) plasmids overnight, plated in Poly-L-Lysine coated 384-well white clear bottom plates in DMEM + 1% dialyzed FBS. Cells were used for assays at a minimum of 6 hr after plating. Culture medium was first removed and cells were stimulated with drugs in assay buffer (1x HBSS, 20 mM HEPES, 1 mg/ml BSA, 0.1 mg/ml ascorbic acid, pH 7.4) for 15 min at room temperature (this and all the following steps), followed by addition of a mixture of isoproterenol (final of 100 nM) and luciferin (final of 1 mM) for $G_{i/o}$-cAMP production inhibition assays and luciferin (final of 1 mM) for $G_s$-cAMP production assays. The plates were counted for luminescence after 25 min in a luminescence plate reader. Results were analyzed using GraphPad Prism 7.0.

## Tango assays

Tango arrestin recruitment assays were carried out according to the previously published procedure *Kroeze et al. (2015)*. In brief, HTLA cells were transiently transfected with receptor TANGO DNA constructs overnight in DMEM with 10% FBS. Transfected cells were then plated into poly-L-Lys coated 384-well plates using DMEM supplemented with 1% dialyzed FBS. After 6 hr incubation, drug dilutions, prepared in DMEM with 1% dFBS, were added for incubation overnight (16–20 hr). Medium and drug solutions were removed, Bright-Glo reagent (20 uL/well) was added for luminescence counting 20 min later. Results were analyzed in GraphPad Prism 7.0.

Bias factors were estimated according to the published procedure *Kenakin et al. (2012)* with modifications. Briefly, normalized and pooled results were analyzed by fitting the Black and Leff operational model in Prism 7.0 to obtain $Log(\tau/K_A)$ values for each pathway (Tango and $G_i$-cAMP). Within each signaling pathway, a difference of $Log(\tau/K_A)$, $\Delta Log(\tau/K_A)$, between a test compound and selected reference (melatonin in this case) was calculated. For a testing compound, the difference of $\Delta Log(\tau/K_A)$, $\Delta\Delta Log(\tau/K_A)$, was then obtained between two pathways. The bias factor is $10^{\Delta\Delta Log(\tau/KA)}$.

## Acknowledgements

We would like to thank the High-Performance Computing Center at the University of Southern California, and the Google Cloud Platform (for higher education and research) for providing computational resources for virtual screening.

## Additional information

### Funding

| Funder | Grant reference number | Author |
|---|---|---|
| National Institute of Diabetes and Digestive and Kidney Diseases | U24DK116195 | Bryan Roth |
| National Institute of Mental Health | RO1MH112205 | Bryan Roth |
| BioXFEL Science and Technology Center | 1231306 | Benjamin Stauch |
| Swedish Research Council | LT000046/2014-L | Linda C Johansson |
| EMBO | ALTF 677-2014 | Benjamin Stauch |

The funders had no role in study design, data collection and interpretation, or the decision to submit the work for publication.

### Author contributions

Nilkanth Patel, Data curation, Software, Formal analysis, Validation, Visualization, Methodology, Writing - original draft, Writing - review and editing; Xi Ping Huang, Resources, Data curation, Formal analysis, Validation, Investigation, Visualization, Writing - review and editing; Jessica M Grandner, Conceptualization, Data curation, Methodology, Writing - review and editing; Linda C Johansson, Validation, Visualization, Writing - review and editing; Benjamin Stauch, Validation, Writing - review and editing; John D McCorvy, Data curation, Formal analysis, Writing - review and editing; Yongfeng Liu, Validation, Investigation; Bryan Roth, Resources, Supervision, Funding acquisition, Writing - review and editing; Vsevolod Katritch, Conceptualization, Resources, Data curation, Software, Supervision, Funding acquisition, Validation, Methodology, Writing - original draft, Project administration, Writing - review and editing

### Author ORCIDs

Nilkanth Patel https://orcid.org/0000-0002-9856-3041
Xi Ping Huang http://orcid.org/0000-0002-2585-653X
Jessica M Grandner https://orcid.org/0000-0001-5068-8665
Linda C Johansson https://orcid.org/0000-0003-4776-5142
Benjamin Stauch http://orcid.org/0000-0001-7626-2021
John D McCorvy https://orcid.org/0000-0001-7555-9413
Vsevolod Katritch https://orcid.org/0000-0003-3883-4505

### Decision letter and Author response

Decision letter https://doi.org/10.7554/eLife.53779.sa1
Author response https://doi.org/10.7554/eLife.53779.sa2

## Additional files

### Supplementary files

• Supplementary file 1. Selected 62 compounds from VLS that were acquired and tested experimentally. The table lists vendors invormation, and Tanimo distances to the closest known MT ligands in ChEMBL database.

- Supplementary file 2. Detailed QC data for the selected 62 compounds.
- Transparent reporting form

## Data availability

All chemical structures of candidate hits and chemical quality control information is deposited as supplementary information. All data generated and analysed during this study is included in the main manuscript or supplementary files.

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
