## [Decision Letter]

**Acceptance summary:**

The recently solved high-resolution structures of two melatonin receptors, MT_1_ and MT_2_, were used for drug discovery. These two GPCRs regulate the circadian rhythm and are well-known targets for the treatment of sleeping disorders (including temporary ones due to jetlag) and mood disorders. The structures were used for virtual screening of small molecules that could bind the receptors and modulate their activities. The results were validated first retrospectively and then also prospectively. The prospective part is particularly interesting because it included the discovery of new hits, some of which have very high activity and selectivity.

**Decision letter after peer review:**

Thank you for submitting your article "Structure-Based Discovery of Potent and Selective Melatonin Receptor Agonists" for consideration by *eLife*. Your article has been reviewed by three peer reviewers, including Nir Ben-Tal as the Reviewing Editors and Reviewer #1, and the evaluation has been overseen by Olga Boudker as the Senior Editor.

The reviewers have discussed the reviews with one another and the Reviewing Editor has drafted this decision to help you prepare a revised submission.

Summary:

Melatonin Receptors are GPCRs that regulate the circadian rhythm, and are used as drug targets for the treatment of sleeping disorders (including temporary ones due to jetlag) and mood disorders. Here, two recently solved crystal structures of Type-1 and 2 Melatonin Receptors (MT) were used to inform a search for more selective MT receptor agonists with novel chemotypes. Virtual ligand screening (VLS) of an 8.4 million compound set helped identifying ten new agonist chemotypes with sub-μM potency, as well as a previously not tested Melatonin derivative. Some of the compounds were selective for one receptor with the most potent new chemotype displaying thirty-fold selectivity for MT_2_ over MT_1_. Correlating the docking poses with functional assays helped investigating the structural basis of signaling bias at the receptors.

Opinion:

This is a carefully conducted and very well timed study that can be published if the issues below are properly handled.

Essential revisions:

1) The docking and functional studies presented are thorough but there are several points that need clarification. "Our benchmarking also underscored the critical role of structural optimization in improving the outcomes….". What was performed is definitely not a "structural optimization" (see also claim in Materials and methods subsection "Receptor Model Preparation and Optimization”) of any kind, but rather structure repair. There was no optimization involved, and the criteria and mode of repair were reasonable, but arbitrary.

a) The authors should explain in the main text if/how the changes introduced in the structure constitute an optimization rather than an arbitrary repair and opening of the ligand-binding pocket for docking. Also, the docking score for selected MT ligands appears to be better than -32 kJ/mol – is that a well-recognized criterion? It seems like an arbitrary number that should be put in the context of, maybe, the docking score of Melatonin.

b) It is quite remarkable that an initial screen yields compounds with such high affinities/potencies: Several compounds were obtained with double-digit nanomolar potencies – properties usually only achieved after several rounds of optimization. Related to that is also the unusual observation that all ligands appear to be agonists – a fact which the authors acknowledge. However, the authors should significantly elaborate on why they think their screen was so successful (in terms of initial hits and compound pharmacology), which would be very valuable to the field of structure-based drug discovery in general.

2) The authors use two different functional assays to test compound activity in two different pathways. However, it isn't always clear which assay/pathway is discussed when the authors list potencies in the main text (particularly in the "Structural basis of Subtype Selectivity of the Hits" section). Also, the G protein signaling assay does not appear to be receptor selective. While it may not be necessary to show these results, what kinds of controls did the authors use in the assay to make sure their response is MT receptor mediated?

3) The authors also investigate the functional selectivity of e.g. compound 37, which shows no G-protein mediated signaling but activates ß-arrestin signaling at MT_1_. The authors suggest that this observation is due to the absence of a hydrogen-bond with N162(4.60) and further conclude that Y281/294(7.38) plays a key role in arresting signaling. However, these suggestions appear speculative and solely rely on the accuracy of the docking pose and previous crystal structures of non-related chemotypes. Instead, the authors need to test structure-guided mutants (e.g. N162Q, Y281F, and others) to examine their hypotheses.

4) The work represents successful use of VLS in laying the groundwork for more selective and potent agonists for MT_1_ and MT_2_. The writing is generally clear and carries the reader through the authors' thoughts and experiments in a logical progression. If anything, the writing is too lean at points. The authors highlight the relevance of the study in the Introduction and Discussion but do not explain the functional differences between MT_1_ and MT_2_. A brief explanation of the role each receptor plays in different biological processes would provide a better context for their intended goal. In addition, how the authors intend to use these results to inform future studies is not clear. What aspects of MT biology/pharmacology do the authors plan to explore? The last sentence of the Discussion just seems like a list of everything Melatonin-related.

5) With drug discovery, aggregation of the putative hits is always a concern. Brian Shoichet wrote a lot about that. It is less likely to be an issue here, e.g., because of the specificity of some of the hits, but we never know. The authors should test that, least for the main hit compounds.

6) It is unclear how close the thermostabilizing mutations are to the binding site (G104A and F251W in MT_1_ or F264W in MT_2_). The authors should highlight these residues in one of the figures and refer to this figure in the relevant places in the text. Because they affect the docking scores it is important to visualize their location relative to the binding site and discuss this in light of point 1a.

7) "To our knowledge, this study represents one of the few cases of successful VLS to identify agonists with novel chemotypes,.….. and a high degree of receptor subtype selectivity for a class A GPCR". This is a rather complex claim for a first, since it involves a specific combination of approaches that are commonly used. The success of the combination can be pointed out, but as always, priority claims are tedious. This may not diminish the importance of the compounds, but it is hardly a trailblazer or surprise of any kind. This should be modified or eliminated.

8) "Thus, it also proves the utility of fragment-like compounds in the early stages of drug discovery." It is not clear why such "proof" was needed, as this is an option in screening because it is considered useful. In any case it can only be a demonstration of successful use, not a proof of anything.

---

## [Author Response]

Essential revisions:1) The docking and functional studies presented are thorough but there are several points that need clarification. "Our benchmarking also underscored the critical role of structural optimization in improving the outcomes….". What was performed is definitely not a "structural optimization" (see also claim in Materials and methods subsection "Receptor Model Preparation and Optimization”) of any kind, but rather structure repair. There was no optimization involved, and the criteria and mode of repair were reasonable, but arbitrary.a) The authors should explain in the main text if/how the changes introduced in the structure constitute an optimization rather than an arbitrary repair and opening of the ligand-binding pocket for docking.

Thank you for pointing to this inaccuracy. In this study we followed our previously published protocol for ligand-guided receptor optimization, “LiBERO” (Katritch et al., 2012). The protocol involves conformational optimization of the pocket side chains and selecting the best resulting conformers in a small scale VLS benchmark with known ligands. Like in several previous examples, this protocol introduces only minor changes in side chain conformation of the receptor binding pocket, resulting in optimized model performance for VLS, as reflected in improved ROC curves shown Figure 1E. Accordingly, we replaced “receptor structural optimization” with more accurate *“ligand guided optimization”* and added the relevant references to the LiBERO method and previous applications of this protocol.

Also, the docking score for selected MT ligands appears to be better than -32 kJ/mol – is that a well-recognized criterion? It seems like an arbitrary number that should be put in the context of, maybe, the docking score of Melatonin.

The value -32 kJ/mol is a standard cutoff value for VLS in ICM, though we agree that it can vary dramatically between receptors, and thus requires further explanation. Here, we introduced the comparative value of these scores by adding the following text: “these scores are better or comparable with the dock score of melatonin (-29.3) and other high affinity ligands of MT receptors”.

b) It is quite remarkable that an initial screen yields compounds with such high affinities/potencies: Several compounds were obtained with double-digit nanomolar potencies – properties usually only achieved after several rounds of optimization. Related to that is also the unusual observation that all ligands appear to be agonists – a fact which the authors acknowledge. However, the authors should significantly elaborate on why they think their screen was so successful (in terms of initial hits and compound pharmacology), which would be very valuable to the field of structure-based drug discovery in general.

We agree that this point requires additional discussion. There are several factors related to both VLS procedure and intrinsic properties of the MT receptor that likely contributed to the high hit rate and agonistic potency of hits. As already mentioned, the high quality of the initial crystal structure, further improved by ligand-guided optimization of the pocket for VLS was important for the success of the screening. Some intrinsic properties of MT receptors also may have contributed to very high potency of hits, for example endogenous ligand melatonin already has exceptionally high potency in single pM range. Melatonin and the other high-potency ligands are small (<250 DA) as they fit small and enclosed MT pocket. Chemical space of such size-limited fragment like libraries is much smaller than usual drug-like space, and can be more exhaustively searched, resulting in better ligands. Regarding agonism preference – yes, most of the known MT receptor ligands are agonists, while antagonists of similar potency are notoriously hard to find, suggesting that agonists are intrinsically preferred ligands for MT receptors. The Discussion section was updated accordingly with these explanations.

2) The authors use two different functional assays to test compound activity in two different pathways. However, it isn't always clear which assay/pathway is discussed when the authors list potencies in the main text (particularly in the "Structural basis of Subtype Selectivity of the Hits" section). Also, the G protein signaling assay does not appear to be receptor selective. While it may not be necessary to show these results, what kinds of controls did the authors use in the assay to make sure their response is MT receptor mediated?

We used in the Gi mediated cAMP inhibition assays in “Structural basis of Subtype Selectivity of Hits”, and updated the text of this section accordingly.

We thank the reviewer for pointing this out. In the G_i/o_-mediated cAMP production inhibition assays with HEK293T cells transiently transfected with MT_1_ or MT_2_ receptor, we also carried out assays at control cells – HEK293T cells without transfections of either MT_1_ or MT_2_ for potential nonspecific activity or activity at endogenous G_i/o_-coupled GPCRs. We didn’t observe any activity at control cells. We have updated the text and specifically stated the observation in Figure 3 legend.

3) The authors also investigate the functional selectivity of e.g. compound 37, which shows no G-protein mediated signaling but activates ß-arrestin signaling at MT1. The authors suggest that this observation is due to the absence of a hydrogen-bond with N162(4.60) and further conclude that Y281/294(7.38) plays a key role in arresting signaling. However, these suggestions appear speculative and solely rely on the accuracy of the docking pose and previous crystal structures of non-related chemotypes. Instead, the authors need to test structure-guided mutants (e.g. N162Q, Y281F, and others) to examine their hypotheses.

We performed functional assays on MT_1_ mutants and showed that compound 37 potency in receptor slightly improved with N162Q mutation, while strongly reduced with Y281F mutation (see Figure 6—figure supplement 1). For comparison, compound 28 drastically (>100-fold) reduced potency in MT_1_ mutants N162Q and Y281F. This new experimental data support our hypothesis that G-protein biased comp 37 lacks a hydrogen bond with N162. We added corresponding text and Figure 6—figure supplement 1.

4) The work represents successful use of VLS in laying the groundwork for more selective and potent agonists for MT1 and MT2. The writing is generally clear and carries the reader through the authors' thoughts and experiments in a logical progression. If anything, the writing is too lean at points. The authors highlight the relevance of the study in the Introduction and Discussion but do not explain the functional differences between MT1 and MT2. A brief explanation of the role each receptor plays in different biological processes would provide a better context for their intended goal. In addition, how the authors intend to use these results to inform future studies is not clear. What aspects of MT biology/pharmacology do the authors plan to explore? The last sentence of the Discussion just seems like a list of everything Melatonin-related.

We thank the reviewer for the suggestion and agree with the notion that MT subtype mediated biology/pharmacology is and will be an important avenue to explore, perhaps with the aid of highly selective MT ligands. We have added/updated the Introduction and Discussion text.

5) With drug discovery, aggregation of the putative hits is always a concern. Brian Shoichet wrote a lot about that. It is less likely to be an issue here, e.g., because of the specificity of some of the hits, but we never know. The authors should test that, least for the main hit compounds.

We agree that ligand non-selective binding and aggregation are the major concerns affecting the validity of the binding and functional assays. To address this concern, we performed off target binding experiments for selected hits (compounds 21, 28 and 37) on a large panel of 47 GPCRs, ion-channels and transporters, and included the results as a new paragraph in text and Figure 6—figure supplement 1. These results indicate that potential aggregation of hits is not a concern for these ligands. Moreover, the result suggest that our top hits may have selectivity towards MT receptors on whole proteome scale, further validating them as new chemical probes.

6) It is unclear how close the thermostabilizing mutations are to the binding site (G104A and F251W in MT1 or F264W in MT2). The authors should highlight these residues in one of the figures and refer to this figure in the relevant places in the text. Because they affect the docking scores it is important to visualize their location relative to the binding site and discuss this in light of point 1a.

The illustration of position of the residues and distances to them are added as Figure 6—figure supplement 2, with corresponding mentions in the main text on p 22. G104A is more than 5 A away and should not make any difference for binding ligand. Similarly, the closest distance for F6.48 residue is 4.3A between hydrophobic residue or receptor and polar amide of the ligand, suggesting lack of significant contact.

7) "To our knowledge, this study represents one of the few cases of successful VLS to identify agonists with novel chemotypes,.….. and a high degree of receptor subtype selectivity for a class A GPCR". This is a rather complex claim for a first, since it involves a specific combination of approaches that are commonly used. The success of the combination can be pointed out, but as always, priority claims are tedious. This may not diminish the importance of the compounds, but it is hardly a trailblazer or surprise of any kind. This should be modified or eliminated.

We agree that such comparison with previous VLS studies are not very adequate here, especially across different receptors (see also answer to comment 1b above), so we removed it.

8) "Thus, it also proves the utility of fragment-like compounds in the early stages of drug discovery." It is not clear why such "proof" was needed, as this is an option in screening because it is considered useful. In any case it can only be a demonstration of successful use, not a proof of anything.

We agree with the reviewers and modified the text accordingly to remove statements of “proof” and “validation” form Discussion.